# Pharmacological Activities of Sulfated Fucose-Rich Polysaccharides after Oral Administration: Perspectives for the Development of New Carbohydrate-Based Drugs

**DOI:** 10.3390/md19080425

**Published:** 2021-07-27

**Authors:** Roberto J. C. Fonseca, Paulo A. S. Mourão

**Affiliations:** 1Laboratório de Tecido Conjuntivo, Hospital Universitário Clementino Fraga Filho, Rio de Janeiro 21941-913, Brazil; pmourao@hucff.ufrj.br; 2Centro de Ciências da Saúde, Instituto de Ciências Biomédicas, Universidade Federal do Rio de Janeiro, Rio de Janeiro 21941-913, Brazil; 3Centro de Ciências da Saúde, Instituto de Bioquímica Médica Leopoldo de Meis, Universidade Federal do Rio de Janeiro, Rio de Janeiro 21941-913, Brazil

**Keywords:** sulfated fucose-rich polysaccharides, sulfated fucan, fucosylated chondroitin sulfate, fucoidan, oral administration, anticoagulant activity

## Abstract

Marine organisms are a source of active biomolecules with immense therapeutic and nutraceutical potential. Sulfated fucose-rich polysaccharides are present in large quantities in these organisms with important pharmacological effects in several biological systems. These polysaccharides include sulfated fucan (as fucoidan) and fucosylated chondroitin sulfate. The development of these polysaccharides as new drugs involves several important steps, among them, demonstration of the effectiveness of these compounds after oral administration. The oral route is the more practical, comfortable and preferred by patients for long-term treatments. In the past 20 years, reports of various pharmacological effects of these polysaccharides orally administered in several animal experimental models and some trials in humans have sparked the possibility for the development of drugs based on sulfated polysaccharides and/or the use of these marine organisms as functional food. This review focuses on the main pharmacological effects of sulfated fucose-rich polysaccharides, with an emphasis on the antidislipidemic, immunomodulatory, antitumor, hypoglycemic and hemostatic effects.

## 1. Introduction

Sulfated fucose-rich polysaccharides have been described in seaweed for approximately a century and are denominated as fucoidan [1]. The structural complexity of these polysaccharides resulted in contradictory reports about their molecular structure since the analytical methods did not allow their detailed characterization. More recently, with the advance of new analytical methodologies, especially high resolution nuclear magnetic resonance, the structure of these polysaccharides has been elucidated [2,3]. Although these studies are restricted to a limited number of species, a high variability is observed among them [4]. Figure 1 shows examples of fucoidans already characterized. One of them contains alternating α (1→3)- and α (1→4)-linked fucose units, while the other is composed exclusively by α (1→3) units (Figure 1a). In both cases, the polysaccharide possesses a heterogeneous sulfation pattern and branches of sulfated and non-sulfated fucose. In addition to fucose, many other sugars are present in these fucoidans, such as galactose, xylose, mannose and uronic acid. It is not possible to clarify whether these sugars are part of the fucoidan molecule or a result of incomplete purification. In terms of the chemical structure, fucoidan could be designated as sulfated fucan (SF). However, “fucoidan” is a traditional denomination and also expresses the heterogeneous composition of these molecules.

In the last 30 years, the study of polysaccharides rich in sulfated fucose was extended to echinoderms (sea urchins and cucumbers) [5,6,7]. In clear contrast with the fucoidans from marine algae, SFs from echinoderms have regular and repetitive structures [8]. Initially, these studies were concentrated on SFs from sea urchins, which are involved in the fertilization process of the invertebrate [9,10]. These SFs are obtained in very small quantities and cannot be tested in in vivo experimental models that require high doses. However, it soon became clear that sea cucumbers also have these SFs, which are present in more expressive quantities. Sea cucumbers contain SFs made up of repetitive tetrasaccharide units, formed by α (1→3) units and with a regular sulfation pattern at positions 2 and 4, which varies among species [11]. Figure 1b shows a representative structure of one of these SFs. In the case of echinoderms, the term SF is appropriate since it denominates chemically homogeneous molecules, distinct from fucoidan.

Moreover, sea cucumbers possess another polysaccharide rich in sulfated fucose, denominated as fucosylated chondroitin sulfate (fucCS) [12]. This compound has a central chain similar to chondroitin sulfate from vertebrates, but it has branches of fucose linked to position 3 of glucuronic acid of the central core. The structure of these fucose branches varies between species. Figure 1c shows the structure of this sulfated polysaccharide. Table 1 and Table 2 show the structural characteristics of fucoidans and fucCS cited in this manuscript. It is important to emphasize that fucoidans can show structural variation according to the source, extraction method and time of the year [13].

These sulfated polysaccharides from echinoderms allowed a significant advance in the attempts to correlate structure and biological activities of these molecules, which is difficult to establish with fucoidans [14]. Many studies report the biological effects of the fucose-rich sulfated polysaccharides administered intravascularly, subcutaneously or intraperitoneally [15,16,17,18,19]. More recently, studies have emerged reporting several pharmacological effects of these polysaccharides after oral administration [20,21]. The observation that these compounds have a therapeutic effect after their oral administration is very significant since it opens the perspective of the use of these molecules for the development of new drugs. The oral route is more practical, comfortable and preferred by patients for long term treatments.

The purpose of this review was to conduct a systematic analysis of the effects observed after oral administration of the sulfated fucose-rich polysaccharides. We distinguished the effects observed with the complex fucoidans from algae from those obtained with the echinoderm polysaccharides. The use of fucCS and SFs is an important pharmacological tool to define structure versus therapeutic effects of polysaccharides rich in fucose as a basis for the development of new drugs.

**Table 1 marinedrugs-19-00425-t001:** Structural characteristics of fucoidans with pharmacological activities after oral administration.

Species	Structure	Sugar and Sulfate Content	Mw (kDa)	Ref
*A. nodosum*	1→3)-α-l-Fucp and a few (1→4)-α-l-Fucp with →3-α-l-(2 and/or 4 Fuc*p*)	The carbohydrate and sulfate content of the fraction A3 were 74.7% and 12.0%, respectively.	97.52	[22]
*S. henslowianum*	→3)-α-l-Fuc*p*(2 SO_3_^−^)-(1→3)-α-l-Fuc*p* (4 SO_3_^−^)-(1→	Fucose and glucose as main sugars. Sulfate content: 25.20%.	ND	[23]
*F. evanescens*	→3)-α-l-Fuc*p*(2 SO_3_^−^)-(1→4)-α-l-Fuc*p*(2 SO_3_^−^)-(1→	Fucose, sulfate and acetyl groups at a molar ratio of 1:1.23:0.36 and trace amounts of galactose and xylose.	10–100	[24]
*F. vesiculosus*	→3)-α-l-Fuc*p*(2 SO_3_^−^)-(1→4)-α-l-Fuc*p*(2,3-SO_3_^−^)-(1→3)	55.9% of carbohydrates, 27.0% of sulfate residues and 5.7% of uronic acid. Carbohydrates were represented mainly by fucose (38%), galactose (3.5%), xylose (2.7%).	20.7	[25]
*C. okamuranus*	→3)-α-l-Fuc*p* (SO_3_^−^)-(1→3)-α-l-Fuc*p*(4 SO_3_^−^)-(1→	The glucuronic acid residues are linked to the C-2 positions of the fucose residues, which are not substituted by a sulfate group. Sulfate content ~15%.	92.1	[26]
*U. pinnatifida*	→3)-α-l-Fuc*p*(2 SO_3_^−^)-(1→4)-α-l-Fuc*p*(2,3-di SO_3_^−^)-(1→3)	This sulphated galactofucan is composed of: galactose 44.6% and fucose 50.9%. Xylose (4.2%), mannose (0.3%). Sulfate content 15%. A significant number of *O*-acetyl groups.	378	[27]
*S. japonica*	→3)-α-l-Fuc*p*(2 SO_3_^−^)-(1→4)-α-l-Fuc*p*(2,3-di SO_3_^−^)-(1→3)	79.49% of fucose and 16.76% of galactose. Sulfate content ~30.72%.	30	[28]
Mozuku *(High molecular weight fraction)*	ND	Sulfate content: 13%.	240	[29]
*L. japonica*	→3)-α-l-Fuc*p*(4 SO_3_^−^)-(1→	46.5% fucoxanthin, 8.01% lipids and 45.4% carbohydrates of mostly cellulose. Sulfate content: 13%.	300	[30]

ND: not determined.

**Table 2 marinedrugs-19-00425-t002:** Structural characteristics of fucCS with pharmacological activities after oral administration.

Species	Proportions of the Branching Sulfated Fucose Units	Mw (kDa)	Ref
*P. graeffei*	81.6% α-Fuc-4SO_4_, 18.4% α-Fuc-2,4diSO_4_	49 kDa	[31]
*I. badionotus*	4.1% α-Fuc-4SO_4_, 95.9% α-Fuc-2,4diSO_4_	70.4 kDa	[32]
*L. grisea*	~27% α-Fuc-2,4diSO_4_; ~20% α-Fuc3,4diSO_4_ and ~53% disaccharides composed of α-Fuc1→2-α-Fuc-3SO_4_→	40 kDa	[20]
*C. frondosa*	The chemical composition contained mainly glucuronic acid, galactosamine and fucose in the molar ratio of 1:1.50:1.16, with 30.07% sulfate content.	14.76 kDa	[33]

## 2. Antidislipidemic Effect

Dyslipidemia refers to a spectrum of metabolic disorders characterized by either an excess or a deficiency of lipoprotein particles, resulting in elevated plasma concentrations of total cholesterol (TC), low-density lipoprotein cholesterol (LDL-C) or triglyceride (TG) and/or depressed high-density lipoprotein cholesterol (HDL-C). Blood levels of different lipoproteins are strongly associated with the risk of cardiovascular diseases [34]. Thus, the management of blood lipid levels has an enormous significance for the control of these diseases. However, many patients fail to reach target levels of lipids with currently available drugs and still experience adverse clinical evolution [35]. Thus, additional pharmaceutical strategies are required to fill these gaps in efficacy and tolerability. There is some research reporting the lipid-lowering effect of sulfated polysaccharide administered by oral route (Table 3).

Initial studies of the antidislipidemic effect of sulfated fucose-rich polysaccharides were performed with sulfated polysaccharide from sea cucumber. A hypolipidemic effect was observed on rats fed with a cholesterol-rich diet and simultaneously received orally SF and fucCS for 6 weeks [36]. The echinoderm polysaccharides significantly decreased TC, LDL-C and the atherogenic index. The authors proposed that these effects may be due to inhibition of Hydroxymethylglutaryl-CoA reductase and/or increased lipoprotein lipase activity, although they did not provide any data regarding this mechanism.

Oral administration of fucoidan from *Ascophyllum nodosum* for 4 weeks improves reverse cholesterol transport in mice [22]. Plasma levels of TC and triglycerides were reduced, as well as fat pad index. The proposed mechanism is related to improvement of the hepatic lipids uptake by activating scavenger receptor B1 and LDL Receptor (LDLR), thus decreasing plasma LDL levels. Another study from the same group showed that oral administration of fucoidan ameliorated atherosclerotic lesion and lipid profiles in a dose-dependent manner in the apolipoprotein (apo) E-deficient mice fed with a high-fat diet [37]. Oil red staining revealed a decrease in the lesion/lumen ratio and in the liver lipid deposition with oral fucoidan-treated apoE^−/^^−^ mice. Moreover, animals treated with the high dose of fucoidan showed reduction of the morphological changes of the kidney induced by high-fat diet. It also reduced triacylglycerol levels and plasma alanine transaminase, suggesting reduction in the high-fat-induced toxicity. Moreover, oral fucoidan increases plasma lipoprotein lipase (LPL) activity, apoA1 and peroxisome proliferator-activated receptor (PPAR) α/β levels. The combination of these effects can improve fatty acid oxidation and lower triglycerides. Another study showed that oral administration of fucoidan from *Sargassum henslowianum* decreased TC, triglyceride and LDL-C levels on obese mice [23].

Subsequent studies attempted to correlate the effect of sulfated fucose-rich polysaccharides on lipid levels with their molecular dynamics using compounds from echinoderms with well-defined chemical structure [38]. FucCS from *Isostichopus badionotus* and SF from *Pearsonothuria graeffei* showed potent effects on triglyceride lowering after oral administration. In contrast, SF from *I. badionotus* showed only weak effects. The distinct effects of these polysaccharides were correlated with their dynamics in solution: fucCS and SF from *P. graeffei* and fucCS from *I. badionotus* form random linear chains in solution with a few spherical aggregations, while SF from *I. badionotus* assumes a spherical conformation in solution and exhibited high viscosity. This study shows a new perspective to explore the structure versus pharmacological effect of sulfated fucose-rich polysaccharides at a molecular level.

**Table 3 marinedrugs-19-00425-t003:** Antidislipidemic effects of sulfated fucose-rich polysaccharides after oral administration.

Polysaccharide	Dosage Regimen and Species	Major Observations and Mechanism Proposed	Ref.
Fucoidan from *A. nodosum*	100 mg/kg/day, 4 weeks, Mice	Improvement of reverse cholesterol transport and bile acid synthesis related genes expression. Reduction of plasma TC (~23.2%) and triglyceride (~48.7%) levels.	[22]
Fucoidan from *A. nodosum*	50 and 100 mg/kg/day, 8 weeks, ApoE^−/−^ mice	Reduction of hepatotoxicity induced by high-fat diet; increased plasma LPL activity, apoA1 level and protein expression of PPARα/β (∼2-fold), improved fatty acid oxidation and TG lowering (∼24.5%).	[37]
Fucoidan from *S. henslowianum*	100 mg/kg/day, 4 weeks, Obese mice	Decreased cholesterol and LDL levels by ~ 23% and 18%, respectively.	[23]
Glycosaminoglycans from *M. scabra*	5, 10, 20 and 50 mg/kg, 6 weeks, Rats	Inhibition of HMG-CoA reductase and/or increased lipoprotein lipase activity and metabolism of cholesterol.	[36]
FucCS and sulfated fucan from *P. graeffei* and from *I. badionotus*	40 mg/kg, 8 days, Rats on high-fat diet	Hypolipidemic activity of sulfated polysaccharides is determined by the molecular dynamics of the sulfated polysaccharide.	[38]

LPL: lipoprotein lipase; PPAR: peroxisome proliferator-activated receptor; TG: triglycerides; HMG-CoA: 3-hidroxi-3-methyl-glutaril-*CoA* reductase. Results obtained with fucoidans from marine brown algae are in blue while those with polysaccharides from echinoderms are in red.

## 3. Anticancer Effect

Cancer is a leading cause of death, along with cardiovascular diseases. The hallmark of cancer treatment has been conventional chemotherapy. Chemotherapeutic drugs target rapidly dividing cells, such as cancer cells; however, these drugs also target normal cells, such as intestinal epithelium, bone marrow and hair follicles. In an attempt to target only cancer cells, a new generation of anticancer drugs arises using specific monoclonal antibodies, small molecule inhibitors and immunotoxins [39]. However, side effects and emerging resistance are still an issue, which increase the demand for new compounds that could act as adjuvant therapy and/or increase efficacy [40]. Regarding this issue, some data in the literature explore the anticancer effect of sulfated polysaccharides after oral administration. Table 4 summarizes the major observations.

An example of the beneficial effect of these polysaccharides as anticancer drugs is the observation that oral doses of fucoidan from *Fucus vesiculosus* delayed tumor growth in a xenograft model and increased cytolytic activity of natural killer cells [41]. Athymic mice were pre-treated with fucoidan daily for 2 weeks and then a human acute promyelocytic leukemia cell line was injected subcutaneously. Significant antitumor activity was observed without any sign of toxicity. Tumor development was clearly slower in the oral fucoidan-treated mice than in the control group. An enhancement of the cytotoxic activity of splenic natural killer cells in mice that were orally treated with fucoidan was also observed, which could be in part responsible for its pharmacological effect. Interestingly, when fucoidan from the same specie were orally administered for 21 days, starting on the seventh day post-tumor implantation, significant reduction in tumor volume and tumor weight was observed when compared with the control group [42]. In vitro assays showed that fucoidan could induce G_0_/G_1_ cell cycle arrest and caspase-dependent apoptosis in diffuse large B-cell lymphoma culture. This indicates that oral fucoidan administration can inhibit tumor growth and development.

Another observation of the anticancer effect of orally administered fucoidan was obtained with Lewis lung carcinoma cells (LLC) [43]. These cells were inoculated into the hypodermic dorsum of mice, and the tumor growth rate was assessed over 21 days. A marked dose-dependent reduction in tumor volume and weight was observed in the fucoidan-treated group, with the maximum effect observed with 144 mg/kg daily oral dose. Expression of growth factors receptors showed a decrease in fucoidan-treated mice compared with the control group. No signs of liver toxicity due to fucoidan administration were observed. Continuous oral administration of fucoidan has a greater efficacy in suppressing tumorigenesis than discontinuous doses, as expected.

In another xenograft model using human prostate carcinoma cells, oral administration of fucoidan for 28 days significantly hindered the tumor growth and tumor vascular density, as indicated by hemoglobin quantification assay [44]. The mRNA expression level of CD31 and CD105, biomarkers of endothelium, also declined. Analysis of the protein expression and gene promoters related to angiogenesis showed that their levels were reduced after oral fucoidan treatment, suggesting that fucoidan hindered tumor growth by inhibiting the formation of new blood vessels.

The anticancer effect of fucoidan was also observed using the polysaccharide from brown alga *Fucus evanescens* using a xenograft model. Colon cancer cells were inoculated into athymic nude mice [24]. Oral treatment with fucoidan for 21 days inhibited tumor growth compared with the vehicle-treated group. The antitumor effect of fucoidan was associated with its inhibition of lymphokine-activated killer T-cell-originated protein kinase (TOPK), highly expressed in many cancers. Tissues from each group were analyzed for phosphorylation of TOPK downstream targets, and the expression of these markers was decreased after 20 days of oral fucoidan treatment. Additional in vitro assays showed that this fucoidan modulates EGF-induced neoplastic transformation of mouse epidermal cells in a concentration-dependent manner. This pathway is related to the machinery that controls fundamental cellular processes, such as growth, proliferation, differentiation, migration and apoptosis. The polysaccharide also binds and decreases TOPK kinase activity in vitro, although a high concentration is required for this effect. The antitumoral activity of the echinoderm polysaccharides has not been tested so far after oral administration. These well-defined structures may help to clarify the effect of sulfated polysaccharides on cancer cells.

**Table 4 marinedrugs-19-00425-t004:** Anticancer effects of sulfated fucose-rich polysaccharides after oral administration.

Polysaccharide	Dosage Regimen and Species	Major Observations and Mechanism Proposed	Ref.
Fucoidan from *F. evanescens*	1–50 mg/ kg, 3 times/week/ up to 21 days, Rats	Inhibition of lymphokine-activated killer T-cell-originated protein kinase (TOPK) (64% at 400 µg/mL) and EGF-downstream signaling. ↓ Tumor growth 72% at 50 mg/kg.	[24]
Fucoidan from *F. vesiculosus*	150 mg/kg/body weight, 2 weeks, Athymic mice	Enhancement of the cytotoxic activity of splenic NK cells (~2.3 fold).	[41]
Fucoidan from *F. vesiculosus*	100 mg/kg, 21 days starting on the seventh day pos tumor implantation, Mice	Induces G_0_/G_1_ cell cycle arrest (2–10%) and caspase-dependent apoptosis.	[42]
Fucoidan from *F. vesiculosus*	144 mg/kg, 26 days, Mice	Reduction of Transforming Growth Factor Receptor (TGFR) levels ( ↓ ~50%) and its downstream signaling pathways. Enhancement of TGFR degradation.	[43]
Fucoidan from *F. vesiculosus*	20 mg/kg, 28 days, Athymic mice	Inhibition of angiogenesis by decreasing mRNA expression level of angiogenesis related markers ( ↓ ~70%) and gene promoters.	[44]

TOPK: Lymphokine-activated killer T-cell-originated protein kinase; EGF: epidermal growth factor; NK: natural killer; TGFR: transforming growth factor receptor. Results obtained with fucoidans from marine brown algae are in blue.

## 4. Immunomodulatory Effect

Immunomodulatory drugs can act at different levels of the immune system. Therefore, different kinds of drugs have been developed that selectively either inhibit or intensify the specific populations of immune responsive cells, i.e., lymphocytes, macrophages, neutrophils, natural killer cells, and cytotoxic T lymphocytes. Immunomodulators affect the cells systems by producing soluble mediators such as cytokines. Therefore, the rational use of drugs with anti-inflammatory effects is necessary to avoid excessive inflammation triggered by external agents or autoimmune diseases and drugs with immunostimulatory effects to increase the immune response such as the production of specific antibodies. In this context, oral administration of sulfated fucose-rich polysaccharides has also shown some interesting effects. A summary of these effects is shown in Table 5.

Oral administration of fucoidan from *Cladosiphon okamuranus* had an antifibrogenesis effect in an N-nitrosodiethylamine-induced liver fibrosis model in rats [26]. Two fractions of fucoidan with distinct molecular weight were tested on this model after oral administration for 12 weeks. A high-molecular-weight fraction of fucoidan prevents liver fibrosis, as indicated by histological examination and hydroxyproline measurement. It also prevents the increase in plasma levels of bilirubin, which occurs as a consequence of liver damage. A low-molecular-weight fraction of fucoidan had only a modest effect on hydroxyproline and bilirubin levels. This observation indicates that the biological effects of fucoidan may differ depending on the molecular weight of the molecule. TGF-β1 appears to play a major role in liver fibrosis and that the mRNA expression of this cytokine is upregulated in this experimental model. The expression of this cytokine decreases significantly in oral fucoidan-treated animals. Furthermore, a chemokine ligand, denominated CXCL12, is markedly stained in the liver epithelium after the induction of experimental fibrosis. Oral treatment with fucoidan prevents the increase of this chemokine expression.

Fucoidan from *F. vesiculosus* was tested on a model of alcohol-induced hepatic disfunction [45]. Seven days of oral administration of this polysaccharide to mice prevents the increase of transaminase levels. It also prevents the expression of TGF-β1 and COX-2, both in the liver from the animal experimental model and in the hepatic cells in culture. Oral fucoidan also decreases mRNA expressions of hepatic inflammatory matrix metalloproteinase-2. Histopathological evaluation showed that macrovesicular steatosis of hepatocytes and focal hepatic necrosis associated with inflammatory cells infiltration induced by high-fat diet was clearly reduced in rats treated with oral fucoidan.

The therapeutic effect of oral fucoidan on non-alcoholic fatty liver disease was tested in rats using an experimental model induced by a high-fat diet [46]. Oral administration of fucoidan for 4 weeks resulted in a decrease of body and liver index and aminotransferase levels when compared with the non-treated group. Total cholesterol and triglycerides also decreased in serum and liver, as well as serum fasting glucose, insulin levels and liver inflammation of the animals fed with fucoidan.

Prevention of arthritis encompasses a variety of the immunomodulatory effects of sulfated polysaccharides. In this particular event, oral administration of fucoidan from *Undaria pinnatifida* for 25 days showed an anti-arthritic effect in a carrageenan-induced paw edema model in rats [27]. Animals treated with fucoidan exhibited significant reduction in paw edema, compared with a standard anti-inflammatory drug, although the doses required to achieve similar protection differ significantly (150 vs 10 mg/kg body weight). Histological analysis revealed that oral fucoidan and standard drug-treated groups exhibited protective effects on joint architecture, such as less edema, cell infiltrations and cartilage destruction. Furthermore, the increase in several biochemical parameters were ameliorated by oral fucoidan administration, and this polysaccharide showed no signs of toxicity at doses up to 1000 mg/kg. Fucoidan demonstrated concentration-dependent antioxidant and anti-inflammatory activities in various in vitro assays, suggesting that its anti-arthritic properties might be related to suppression of prostaglandin production and other inflammatory mediators.

Another study showed that oral administration of fucoidan from *F. vesiculosus* for 13 weeks to Goto-Kakizaki rats, which spontaneously develop mild hyperglycemia and hyperinsulinemia, protected the animals from diabetes nephropathy [47]. The increased fasting blood glucose, urea, serum creatinine and urine protein levels observed in positive control animals were significantly decreased in GK rats that received fucoidan orally at both doses. Fucoidan diminished levels of collagen IV in the renal cortex and decreased expression of TGF-β1 and fibronectin in the renal cortex and in the glomerular mesangial cells. Histopathological analysis revealed vacuolation of renal tubular epithelial cells and inflammatory cell infiltration in the renal interstitium of the kidneys from the diabetic rats compared with those from the control and fucoidan-treated rats. The increased expression of NF-κB in the nuclei of glomerular mesangial cells was also attenuated significantly by the oral administration of fucoidan, suggesting that this pathway is involved in the nephropathy and in the anti-inflammatory activity of the polysaccharide.

Suppression of allergic symptoms is another event related to immunomodulatory activity of fucoidan [28]. The effect of the polysaccharide in this particular event was investigated using fucoidan from *S. japonica* orally administered for 4 days to mice submitted to a passive cutaneous anaphylaxis reaction. The ear edema was evaluated 2 h after antigen challenge. Fucoidan showed an inhibitory effect only after oral but not intraperitoneal administration. The mechanism proposed is related to an increase in galectin-9 expression in intestinal epithelial cells and in the blood of mice fed with fucoidan. In fact, administration of anti-galectin 9 antibody suppressed fucoidan effects on ear edema. Moreover, this polysaccharide prevented the interaction of IgE and mast cells, an important event that mediates allergic responses. These data suggest that dietary intake of fucoidan from *Saccharina japonica* may prevent allergic symptoms.

The impact of the molecular weight on the anti-inflammatory activity of sulfated fucan from the sea cucumber *Acaudina molpadioides* with varying degrees of polymerization was also reported [48]. The SF was tested on an animal model of intestinal mucositis after oral administration for 26 days. Histological analysis revealed that morphology of the intestinal mucosa of fucoidan-treated animals was similar to the healthy group and that this effect was more pronounced with the high-molecular-weight fractions. Interestingly, these fractions regulated Th1/Th2 immune balance processes by altering IFN-γ/IL-4 ratio, while the oral administration of intact fucoidan had no effect. Intact SF and the high-molecular-weight fractions enhanced IgA protein expression levels in intestinal mucosa and strengthened intestinal adaptive immunity. Another interesting aspect of this work was the analysis of plasma concentration achieved by the oral administration of the SF. The low-molecular-weight fractions achieved high plasma levels when compared with unfractioned polysaccharide; therefore, the absorption and bioavailability of SF are likely to depend on the molecular size of the polysaccharide.

Another approach investigated the anti-inflammatory effect of oral administration of fucCS from the sea cucumber *I. badionotus* for 7 days. In contrast with the highly heterogeneous fucoidan from brown algae, this polysaccharide has a regular repetitive structure containing mostly 2,4 disulfated fucose units, as shown in Figure 1c. When tested on an experimental model of colitis induced by dextran sulfate, oral fucCS attenuated the body weight loss, expression of colonic TNF-α gene and colon shortening caused by experimental colitis. The authors proposed that this protective effect might be due to downregulation of NF-kB and downstream genes such as COX-2 and TNF-α and a benefic profile on gut microbiota [49].

**Table 5 marinedrugs-19-00425-t005:** Immunomodulatory effects of sulfated fucose-rich polysaccharides after oral administration.

Immunomodulatory Effect	Polysaccharide	Dosage Regimen and Species	Major Observations and Mechanism Proposed	Ref.
Antifibrotic effect	Fucoidan from *C. okamuranus*	Free access to drinking water containing 2% low (28.8 kDa) or high (41.4 kDa) MW fractions, 12 weeks, Rats	↓ TGF-β1 mRNA expression and the levels of chemokine ligand CXCL12 in the liver (~3 fold).	[26]
Hepatoprotection	Fucoidan from *F. vesiculosus*	30 or 60 mg/kg, 7 days, Mice	↓ expression of liver TGF-β1(~40%) and COX-2, ↑ antioxidant pathways.	[45]
Hepatoprotection	Fucoidan from *F. vesiculosus*	100 mg/kg, 4 weeks, Rats on high-fat diet	↓ TNF-α, IL-1β and MMP-2 mRNA expressions (~50–70%). Prevention of the increase in serum lipids and glucose levels induced by HFD.	[46]
Nephroprotection	Fucoidan from *F. vesiculosus*	50 and 75 mg/kg, 13 weeks, Rats	Decreased levels of collagen IV, NF-κB, TGF-β1 and fibronectin in the renal cortex and in the glomerular mesangial cells.	[47]
Anti-arthritic and antioxidant effects	Fucoidan from *U. pinnatifida*	50 or 150 mg/kg, 25 days, Rats	Downregulation of COX-2 and other inflammatory mediators (68% inhibition of in vivo inflammation).	[27]
Immunostimulatory effects	Fucoidan from *U. pinnatifida*	300 mg daily,20 weeks, Human	Higher immunogenicity of influenza trivalent vaccine than control group and increase of natural killer cell activity.	[50]
Suppression of allergic symptoms	Fucoidan from *S. japonica*	100–400 μg/day,4 days, Rats	Prevention of the interaction of IgE and mast cells via an increase in galectin-9 mRNA expression ( ↑ ~50%) in intestinal epithelial cells.	[28]
Anti-inflammatory effect	FucCS from *I. badionotus*	80 m/kg, 7 days, Rats	Downregulation of NF-kB and downstream genes such as COX-2 and TNF-α and a benefic effect on gut microbiota.	[49]
Anti-inflammatory effect	Sulfated fucan from *A. molpadioides* with varying degrees of polymerization (10–500 kDa)	50 mg/kg, 26 days, Mice	Regulation of IFN-γ/IL-4 ratio (0.53 to 0.70) and Th1/Th2 response, IL-6 and IL-10 levels, enhanced IgA protein expression levels (~35%) in intestinal mucosa.	[48]

CXCL12: C-X-C motif chemokine ligand 12; TNF-α: tumor necrosis factor; TGF-β: transforming growth factor beta; NF-kB: nuclear factor kappa B; COX-2: ciclooxigenase 2; IFN-γ: interferon gamma; IgA: immunoglobulin A; HFD: high-fat diet. Results obtained with fucoidans from marine brown algae are in blue while those with polysaccharides from echinoderms are in red.

## 5. Effects on Diabetes

Diabetes is a highly prevalent disease characterized by high levels of blood sugar, due to deficiency of insulin concentration and/or activity. Pharmacological therapy may be required in order to maintain normal level of blood glucose and to delay or prevent the development of diabetes-related health problems. The first choice in type 2 diabetes is oral hypoglycemic drugs, but side effects, toxicity and unwanted drug–drug interactions can compromise the effectiveness of the treatment [51]. Nevertheless, the idea of a diet rich in sulfated fucose-rich polysaccharides with hypoglycemic effect as adjuvant therapy may be an interesting alternative. A summary of these effects is shown in Table 6.

Oral administration of fucoidan from *F. vesiculosus* for 13 weeks to Goto-Kakizaki rats reduced high blood glucose and recovers serum insulin levels [52]. Moreover, histopathological analysis of the pancreas also demonstrated that fucoidan markedly reduced islet atrophy, fibrosis and inflammation. Additional in vitro assays showed that treatment with the phosphodiesterase inhibitor significantly increased fucoidan-induced insulin secretion, whereas treatment with the adenylyl cyclase inhibitor significantly decreased fucoidan-induced insulin secretion. These results suggested that the cAMP signaling pathway may be important in the antidiabetic effect of fucoidan. A further study showed that the polysaccharide inhibits dipeptidyl peptidase-IV, which prolongs the action of incretins, reduces glucose and increases insulin production. This is another possible mechanism involved in the antihyperglycemic effect of fucoidan [25].

A detailed study about the effect of oral sulfated fucose-rich polysaccharides on diabetes employed an experimental model in mice inducing type 2 diabetes by high fat/sucrose diet [53]. The authors tested a fucCS from the sea cucumber *Cucumaria frondose*. Oral administration for 19 weeks stimulated insulin-dependent glucose uptake in skeletal muscle cells and improved insulin sensitivity. Oral fucCS treatment promoted insulin-stimulated phosphorylation of phosphoinositide 3-kinase and protein kinase B, the major regulators of glucose uptake response to insulin in skeletal muscle and increased GLUT4 translocation. It also increased mRNA expression levels of these regulators in the skeletal muscle of oral polysaccharide-treated mice. Furthermore, fucCS increased hepatic glycogen content and restored the activities of key enzymes for glucose metabolism in the liver to near-control levels [54]. Therefore, oral fucCS can promote hepatic glycogen synthesis by regulating gene expression.

A further study attempted to investigate the mechanisms involved in the favorable effect of fucCS on experimental diabetes [33]. Animals were submitted to a high-fat/sucrose diet, which disrupts insulin signaling and thus results in endoplasmic reticulum stress and inflammation. After oral administration of oral fucCS for 19 weeks, several cytokines and inflammatory markers were reduced in the serum and in the liver of treated animals. Analysis of mRNA expression showed that the polysaccharide attenuates the increase of several markers of liver endoplasmic reticulum stress, inhibits important inflammatory signaling pathways and improves insulin sensitivity in the liver.

The antidiabetic effect of sulfated fucose-rich polysaccharides extracted from 10 low-edible-value sea cucumber species was tested after oral administration for 8 weeks using a classic experimental model of diabetes induced by streptozotocin in rats [32]. A variety of effects were observed, such as reduced polyphagia and loss of body weight, decreased fasting blood glucose level and improved glucose tolerance by increasing insulin secretion and enhancement of its sensitivity. A significant improvement of antioxidant enzymes was also observed indicating a decrease in inflammatory status and oxidative stress. The sulfated polysaccharides decrease the levels of transaminases, suggesting a repair of liver damage associated with the experimental model. They also restored normal levels of TNF-α content in the serum and enhanced synthesis of liver glycogen to decrease blood glucose level. Furthermore, they reduced levels of serum triacylglycerol, TC and LDL-C and increased HDL-C/LDL-C values, which indicates that oral administration of sulfated fucose-rich polysaccharides can alleviate dyslipidemia resulting from diabetes. In this study, the authors did not show a clear correlation between the structure of the polysaccharide and its biological effect. Nevertheless, the sulfated polysaccharides from *C. frondosa* and *Thelenota ananás* seem to show more potent effects.

**Table 6 marinedrugs-19-00425-t006:** Hypoglycemic effects of sulfated fucose-rich polysaccharides after oral administration.

Polysaccharide	Dosage Regimen and Species	Major Observations and Mechanism Proposed	Ref.
Fucoidan from *F. vesiculosus*	75 m/kg, 13 weeks, Rats	Reduced islet atrophy, fibrosis and inflammation mediated by cAMP signaling pathway. Inhibition of dipeptidyl peptidase-IV.	[25,52]
High molecular weight fucoidan from Mozuku *(C. okamuranus)*	1620 mg, 12 weeks, Human	Alterations in GLP-1 (from 6.42 ± 3.52 to 4.93 ± 1.88 pmol/L) and hemoglobin A1c levels (from 6.73 ± 1.00 to 6.59 ± 1.00).	[29]
Fucoidan extract from *Laminaria* ssp.	500 mg, 3 months, Human	Decrease in diastolic blood pressure and LDL-C (↓13%) with increase in insulin levels (↑ 30%), HOMA β-cell, and HOMA IR.	[29,55]
FucCS from *C. frondosa*	20 or 80 mg/kg, 19 weeks, Mice	↑ insulin-stimulated phosphorylation of PI3K and PKB; ↑ GLUT4 translocation ↑ glycogen synthesis-related gene expression; ↓ liver ER stress markers, ROS, TNF-α and other inflammatory markers levels in serum and liver; ↓ inflammatory signaling pathways in the liver.	[33,53,54]
Sulfated polysaccharides from 10 sea cucumber species	200 or 400 mg/kg, 8 weeks, Rats	↓ TNF-α, ↑ antioxidant enzymes; ↑ glucose metabolism related gene signaling pathway.	[32]

GLP-1: glucagon-like peptide 1; PI3K: phosphatidylinositol 3-kinase; PKB: protein kinase B; GLUT4: glucose transporter 4; ER: endoplasmic reticulum; ROS: reactive oxygen species. Results obtained with fucoidans from marine brown algae are in blue while those with polysaccharides from echinoderms are in red.

## 6. Thrombosis and Hemostasis

Thromboembolic events are expanding due to the aging of the population and a more precise diagnosis. Heparin is the classic anticoagulant used in the treatment and prevention of thrombosis, but its use is limited to the intravenous or subcutaneous route, and it has significant adverse effects [56,57,58]. New oral anticoagulants are available, but bleeding is still a concern [59]. Therefore, there is a demand for new antithrombotic drugs.

The antithrombotic effects were the first significant pharmacological effects reported for the sulfated fucose-rich polysaccharides [60,61,62]. Several authors addressed the parenteral use of fucoidan and echinoderm polysaccharides in experimental models of venous and arterial thrombosis [63,64,65]. The initial studies associate the mechanism of action of these molecules with heparin, the most traditional anticoagulant sulfated polysaccharide. However, recent studies using sea cucumber fucCS showed that the anticoagulant mechanism of this compound is serpin-independent, inhibiting the assembly of the tenase and prothrombinase complexes and the generation of thrombin and factor Xa [66]. In addition to the distinct mechanism of action, the preserved antithrombotic effect after oral administration has made this sulfated polysaccharide an interesting candidate for the development of new drugs [20,67]. Table 7 summarizes the effects of sulfated fucose-rich polysaccharide in hemostasis.

An initial study about the antithrombotic effect of oral fucoidan employed a low-molecular-weight fraction obtained by chemical degradation of the native polysaccharide from *L. japonica* [68]. After oral administration for 30 days to rats, the polysaccharide prolonged aPTT and TT values, increased TFPI and suppressed thromboxane levels in rat plasma. It also inhibited thrombin-induced platelet aggregation and enhanced fibrinolysis. The antithrombotic effect was tested in an arterial thrombosis model induced by electrical stimulus. The low-molecular-weight fucoidan prolonged the time for formation of the thrombus. Unlike aspirin, the low-molecular-weight fucoidan did not decrease platelet number and fibrinogen level after oral administration for 30 days, which suggest a safe antithrombotic profile.

The first report of the antithrombotic effect of an echinoderm polysaccharide after oral administration employed a fucCS from the sea cucumber *Ludwigothurea grisea* [20]. The polysaccharide increased aPTT and TT values and decreased thrombin residual activity. A dose-dependent antithrombotic effect is observed using a vena cava and an arterial shunt thrombosis models in rats. After removal of the fucose branches, the antithrombotic activity of the polysaccharide was abolished. The dose necessary to achieve complete inhibition of the thrombus formation was 50 mg/kg administered in aqueous solution. A great achievement was the encapsulation of the polysaccharide on gastro-resistant tablets, which prevents the degradation in the acid juice fluid [67]. This approach allowed the dose of fucCS to be decreased to 25 mg/kg and to still observe the same anticoagulant and antithrombotic effects. FucCS does not alter bleeding tendency or arterial pressure after oral administration, which is the major concern with this polysaccharide due to activation of the contact system and the release of bradykinin [69].

Recently, oligosaccharides containing 6→18 units were obtained by controlled depolymerization of fucCS from the sea cucumber *P. graeffei* [70]. These oligosaccharides were the active ingredient of gastro-resistant microcapsules using a chitosan-coated alginate system and were orally administered to rats in a single dose of 10 or 50 mg/kg. Microcapsules containing the oligosaccharides prolonged aPTT values with a stronger intensity compared with microcapsules containing native fucCS. In a venous thrombosis model, oral administration of 50 mg/kg fucCS oligomers delivered by aqueous solution exhibited a weaker antithrombotic effect than observed with gastro-resistant microcapsules, probably due to the partial removal of sulfated fucose branches in the acid gastric fluid. No bleeding tendency was observed for fucCS oligomers tested on gastro-resistant microcapsules. Using an intestinal Caco-2 cell system, the authors confirmed that fucCS oligomers showed higher absorption than native polysaccharide.

A very curious observation comes from a study involving fucoidan from *F. vesiculosus* and *Laminaria japonica* orally administered twice daily in a multiweek dose-escalation study of dogs with hemophilia A [71]. A dose-dependent decrease in bleeding time score and improved clotting dynamics was observed, indicating a procoagulant effect of these polysaccharides after oral administration. In vitro assays showed that this fucoidan inhibited exogenous TFPI activity and accelerated the clotting time of human hemophilia A and B plasma. Current methods of hemophilia treatment are expensive, challenging and involve regular administration of clotting factors. While gene therapy is expensive and still under investigation, additional therapeutic options have already explored heparin-like sulfated polysaccharides, including pentosan polysulfate and fucoidan, with unique procoagulant activity for bleeding disorders [72]. These results explore another aspect of the effects of sulfated polysaccharides on the coagulation system. Interestingly, sulfated polysaccharides from red algae have already shown a dual effect on coagulation either as a pro- or anticoagulant drug [73].

**Table 7 marinedrugs-19-00425-t007:** Effects on hemostasis of sulfated fucose-rich polysaccharides after oral administration.

Polysaccharide	Dosage Regimen and Species	Major Observations and Mechanism Proposed	Ref.
Low molecular weight fucoidan (Mw7.6 kDa) from *L. japonica*	400 and 800 mg/kg 30 days, Rats	↑ TFPI (4.5 to 110.2 U/mL) and 6-keto-PGF1α levels (32.8 to 50.4 U/mL). ↑ Fibrinolysis (tPA and PAI-1 levels) ↓ Thromboxane A2 levels.	[68]
Fucoidan from *L. japonica*	400 mg for 5 weeks to humans	↑ 6-keto-PGF1a (44 to 113 ng/L) ↑ fibrinolysis.	[30]
Fucoidans from *F. vesiculosus* and *L. japonica*	5–20 mg/kg, Twice daily in a multiweek escalation dose, Dogs	Procoagulant effect, Inhibition of TFPI activity.	[71]
Native and gastro-resistant tablets of fucCS from *L. grisea*	5–50 mg/kg, Single dose or 5 days, Rats	Serpin-independent anticoagulant effect by inhibiting the formation of factor Xa and/or IIa through the procoagulants tenase and prothrombinase complexes. Antithrombotic effects at 50 mg/kg: ~85% vs. 55% inhibition of the venous and arterial thrombus weight, respectively.	[20,66,67]
Gastro-resistant tablets containing FucCS oligomers (6 to 18 saccharide units, Mw 3,4 kDa) from *P. graeffei*	10 or 50 mg/kg Single dose, Rats	Anticoagulant and antithrombotic effects (82% of venous thrombosis inhibition at 50 mg/kg).	[70]

TFPI: tissue factor pathway inhibitor; *6*-*keto* PGF1α*: 6*-*keto* prostaglandin F1α; tPA: tissue plasminogen activator; PAI-1: plasminogen activator inhibitor. Results obtained with fucoidans from marine brown algae are in blue while those with polysaccharides from echinoderms are in red.

## 7. Clinical Trials

Preclinical studies using animal models are important to assess the effectiveness of fucose-rich polysaccharides in different pathologies and to elucidate the mechanisms involved in their mechanism of action. Clinical trials are the next step for the development of these polysaccharides as new drugs and/or using marine organisms as a food supplement, and some studies in the literature address this issue. These aspects are also under investigation, and we describe the major observations in this review.

Very few studies report the anticancer effect of fucoidan in humans. Some clinical trials report an improvement in the quality of life of patients who have used fucoidan orally as an adjuvant therapy. Cancer patients receiving oral fucoidan for 4 weeks showed reduced levels of proinflammatory cytokines, including IL-1β, IL-6 and TNF-α [74]. Interestingly, the responsiveness of IL-1β was significantly correlated with overall survival, suggesting that this might be a useful prognostic biomarker for advanced cancer patients receiving fucoidan. Another study examined the effects of fucoidan extracted from *C. okamuranus* on natural killer cell activity in cancer survivors [75]. Male patients treated with 3 g of fucoidan for 6 months showed an enhanced activation of natural killer cells. Fucoidan also has the potential for adjuvant therapy and may also reduce chemotherapy toxicity for cancer patients [76,77].

Another study in humans involves the immunogenicity response to influenza trivalent vaccine in the elderly, whose antibody production is generally attenuated [50]. Oral intake of fucoidan from seaweed *U. pinnatifida* for 20 weeks (300 mg daily) increased antibody titers, which is most evident against influenza B strain. This effect may be related to NK cell activity. This suggest that popular seaweeds containing fucoidan that are eaten daily in Japan could have immunostimulatory effects in enhancing vaccination efficacy. Another study showed no decrease of osteoarthritis symptoms after a 300 mg daily oral dose of *F. vesiculosus* extract (85% fucoidan) over a 12-week period [78].

Studies of the effect of sulfated polysaccharides on diabetes after oral administration were also reported in humans [29]. Thirty patients with type 2 diabetes were selected for oral intake of a high-molecular-weight fucoidan from Mozuku seaweed for a 12-week period. Oral fucoidan altered hemoglobin A1c and levels of glucagon-like peptide-1 and increased the number of bowel movements and stool frequency. These effects were associated with a beneficial control of diabetes. Another randomized, double-blind, placebo-controlled clinical trial was carried out with 25 overweight volunteers to evaluate the effect of fucoidan administration on insulin secretion and sensitivity [55]. A total of 13 patients received an oral dose of 500 mg of fucoidan once daily before breakfast and 12 patients received placebo for 3 months. A significant decrease in diastolic blood pressure and LDL levels with an increase in insulin levels were observed after oral fucoidan administration. There were no significant adverse events associated with the long-term intake of fucoidan in both studies.

Human studies reporting the effect of sulfated polysaccharides on hemostasis are scarce, as in the case of other biological effect. In one study, oral administration of capsules containing 400 mg fucoidan from *L. japonica* to healthy participants for 5 weeks resulted in increased fibrinolysis and antiplatelet effects [30]. Fucoidan was not detected in the plasma, probably due to low polysaccharide concentrations and/or the sensitivity of the method used. This is one of the challenges associated with assessing the pharmacokinetics of orally administered sulfated polysaccharides.

## 8. Future Perspectives: Pharmacokinetics Studies and Prebiotic Effects

This review summarizes the therapeutic effects achieved after oral administration of sulfated polysaccharides in a variety of pathological processes. These observations were obtained mainly using animal experimental models, although some preliminary data have already been reported in humans. These results are not limited to the therapeutic effect but also highlight the proposed molecular mechanisms involved in the pharmacological action of these polysaccharides. Further studies are necessary to further understand their pharmacokinetic and the modulating effect on the intestinal microbiota.

In the case of heparin, a paradigm of an anticoagulant drug with carbohydrate structure, the transition from intravenous to subcutaneous administration was associated with the development of low-molecular-weight heparin. This led to the development of new analytical methods to study its pharmacodynamics, resulting in the now widespread methods to determine the plasma concentration of heparins based on anti-FXa and anti-FIIa assays [56]. Likewise, there is a need to develop sensitive methods for the study of the pharmacokinetics/pharmacodynamics of fucose-rich polysaccharides after their oral administration.

A similar approach was employed for oral administration of fucCS. The plasma concentration of the polysaccharide was determined using ex vivo coagulation assays and purified proteases [67]. The results allowed the correlation of the anticoagulant with its antithrombotic effects. However, it was not possible to evaluate tissue distribution, minor structural modifications and urinary elimination of the compound. Such analyses are limited by the very low concentration achieved by the polysaccharide after oral administration. We need to validate new methods for fucCS labeling and quantification as a critical step for the pharmacokinetic studies.

A partially depolymerized fucCS, administered as a single oral dose of 50 mg/kg, was detected in plasma between 0.5 and 7.5 h using a chromatographic method. Only 0.1% of the dose was detected in the urine accumulated during 24 h [79]. It is very challenging to measure the plasma levels of these polysaccharides analytically because of the heterogeneous molecular weight, branched structure and similarity in monosaccharide composition to mammalian polysaccharides.

Some studies report other sensitive methods to assess the pharmacodynamics of fucose-rich polysaccharides after oral administration. In one study, the authors employed an antibody against fucoidan extracted from *C. okamuranus* and developed a sensitive ELISA method for the measurement of its serum and urinary concentration after a single oral dose of fucoidan (1 g) in ten healthy volunteers [80]. The anti-fucoidan antibody specifically recognized fucoidan from *C. okamuranus* and *F. vesiculosus* with different specificities, with low cross-reactivity with heparin and heparin-like substances. Fucoidan concentration in serum and urine was detectable 3 h and mostly 6 h after its administration. The time and peak concentrations varied among individuals, suggesting a high variability of fucoidan absorption in the intestine. The concentration of fucoidan was higher in the urine than in the serum. The molecular weight of the ingested fucoidan remained unchanged in the serum, whereas the fucoidan excreted in the urine showed an expressive decrease in size. Possibly, fucoidan degradation occurs mostly in the excretory system but not during its absorption through the gastrointestinal tract by local microbiota. Using the same method, a further study confirmed fucoidan in the urine of Japanese volunteers after 100 g of oral intake of seaweed *C. okamuranus* [81].

Another work using fucoidan antibody revealed that the polysaccharide accumulated in jejunal epithelial cells, mononuclear cells in the jejunal lamina propria and sinusoidal non-parenchymal cells in the liver of rats fed standard chow containing 2% fucoidan for one or two weeks [82]. Fucoidan was detected in the sinusoids of hepatic lobules, which suggested its internalization by macrophages. The intestinal absorption was also observed using an intestinal Caco-2 cells system in vitro.

One of the major challenges associated with assessing the bioavailability of orally administered fucoidan has been the lack of a sensitive and accurate analytical method that can quantify fucoidans in the blood since this polysaccharide exhibits low anti-FXa and anti-FIIa activities compared with heparin. However, one study evaluated the pharmacokinetics and tissue distribution of fucoidan in rats after a single-dose oral administration of 100 mg/kg of fucoidan from *F. vesiculosus* based on its anti-Xa activity. The C_max_ in plasma was observed at 4 h after oral administration. Fucoidan accumulated mainly in the kidney and was also present in liver and spleen and showed a relatively long absorption time and extended circulation in the blood [83]. Different analytical methods are reported for the evaluation of pharmacokinetic parameters of marine-derived drugs [84].

The classic mechanism of action proposed for the sulfated fucose-rich polysaccharides after oral administration is summarized in Figure 2A. The polysaccharides are absorbed through the gastrointestinal tract, probably by endocytosis due to their high molecular weights [85], reach appropriate plasma concentration and exert their therapeutic action. Subsequently, the polysaccharides are distributed to different tissues, metabolized and excreted. Structural modifications might occur during these processes.

Moreover, there is evidence for another mechanism involved in the therapeutic effect of fucose-rich sulfated polysaccharides administered orally, which involves modification of the gut microbiota induced by the polysaccharides [86]. Probiotics are important microorganisms in the intestinal microflora. When colonized in adequate amounts, they confer a health benefit to the host by modulating several physiological activities [87]. Gut microbiota degrade polysaccharides and produce short-chain fatty acids, which might play an important role in maintaining the epithelial barrier function, regulating the immune responses and metabolic processes as well as inhibiting tumor development [88,89,90,91]. Models of gut microbe cultivation in vitro provide a convenient way to study the structural modifications of polysaccharides during digestion and absorption in the gastrointestinal tract and have already demonstrated a large number of applications in the field of intestinal fermentation of polysaccharides and oligosaccharides [92,93].

This particular aspect was investigated using a fucCS from the sea cucumber *P. graefei* [31]. The incubation of the polysaccharide with human intestinal flora in a simulated intestinal digestion model in vitro induced changes of intestinal microflora and degradation of the polysaccharide. Three samples of these bacteria utilize fucCS as a carbon source for their growth and produced short-chain fatty acids that decreased the pH of the media. A high content of acetate, propionate and butyrate was observed, indicating that they were the major products of microbial metabolism. Propionic acid can inhibit cholesterol synthesis in the liver, promote redistribution of cholesterol in plasma and liver, inhibit lipogenesis enzymes and reduce plasma lipid levels [94]. This activity could be responsible, in part, for the hypolipidemic effect of fucCS after oral administration. *Bifidobacterium, Bacteriodes prevotella* and three species of *Clostridium* seemed to be involved in the metabolism of fucCS. These observations are summarized in Figure 2B.

Another study reported that fucCS from *S. japonicas* was not broken down under salivary and gastrointestinal digestion [95]. Due to the inhibition of pancreatic lipase in vitro by increasing concentrations of fucCS, the authors hypothesized that fucCS may work at the level of the gastrointestinal tract itself and/or after its absorption for the hypolipidemic effect.

The intestinal flora of part of the human population can ferment fucoidan to afford low-molecular-weight oligosaccharides [96], while another research had a different conclusion [97]. This suggests that the consumption of non-sterile marine foods with associated bacteria may have been the route by which these novel enzymes were acquired in human gut. Interestingly, the consumption of *C. okamuranus* algae by Japanese volunteers was associated with increased oral absorption of fucoidan contained in algae [98,99]; therefore, these contradictory results may be due to individual differences between species and strain level, which results in different metabolic capabilities of the microbiota to hydrolyze the molecules. Obviously, it also may depend on the polysaccharide structure. Further research is necessary to assess whether these changes in both bacterial composition and sulfated polysaccharide degradation also occur in vivo, and critically, whether such changes are responsible for some of the biological effects of these sulfated fucose-rich polysaccharides after oral administration.

## 9. Conclusions

Sulfated fucose-rich polysaccharides from marine organisms are unique molecules with various pharmacological effects. They might have promising therapeutic applications in different diseases. There has been an increasing interest in the therapeutic use of natural products for treatment of chronic cardiovascular and/or inflammatory diseases. The fact that these sulfated polysaccharides preserve their pharmacological effect after oral administration opens the perspective for the development of new drugs and/or the use of marine organisms as a source of functional food. High doses of the orally administered sulfated polysaccharides from marine organisms are employed in most studies. This limits their therapeutic use. Further larger trials are required to establish the role of fucoidan in several diseases. More efficient techniques of labeling sulfated polysaccharides will help to understand the pharmacokinetic parameters of these molecules. In recent years, much attention has been given to the prebiotic function of bioactive polysaccharides. Elucidating the linkage between the sulfated polysaccharides and gut microbiota will help to understand the biological effects of these molecules after oral administration.

## Figures and Tables

**Figure 1 marinedrugs-19-00425-f001:**
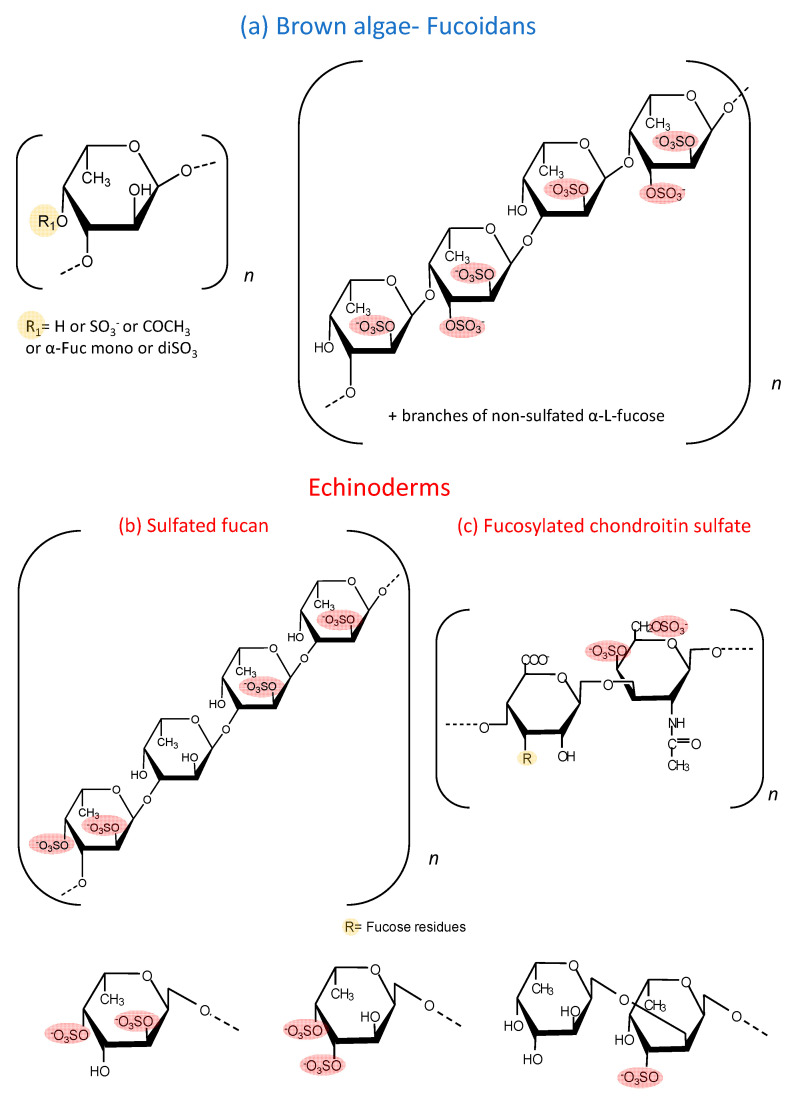
Structure of the sulfated fucose-rich polysaccharides from brown algae and echinoderms. (**a**) Fucoidans from brown algae are composed of α (1→3)-linked fucose units or alternating α (1→3)- and α (1→4)-linked fucose units. Mannose, galactose, xylose, uronic acid and branches of other monosaccharides make this polysaccharide highly variable and with complex structures. (**b**) SFs from echinoderms are made up of a repetitive tetrasaccharide units, formed by α (1→3) units and with a regular sulfation pattern at positions 2 and 4. (**c**) Structure of a fucCS from sea cucumbers. This polysaccharide has a chondroitin sulfate-like backbone, with branches of α-fucose linked to position 3 of the β-glucuronic acid of the central core. These fucose branches varies among species. In the specie *L. grisea*, for example, three types of branches are observed: α-Fuc-2,4diSO_4_, α-Fuc-3,4diSO_4_ and disaccharides composed of α-Fuc1→2-α-Fuc-3SO_4_→. The sulfated fucose-rich polysaccharides from echinoderms have a more regular and repetitive structures compared with brown algae polysaccharides.

**Figure 2 marinedrugs-19-00425-f002:**
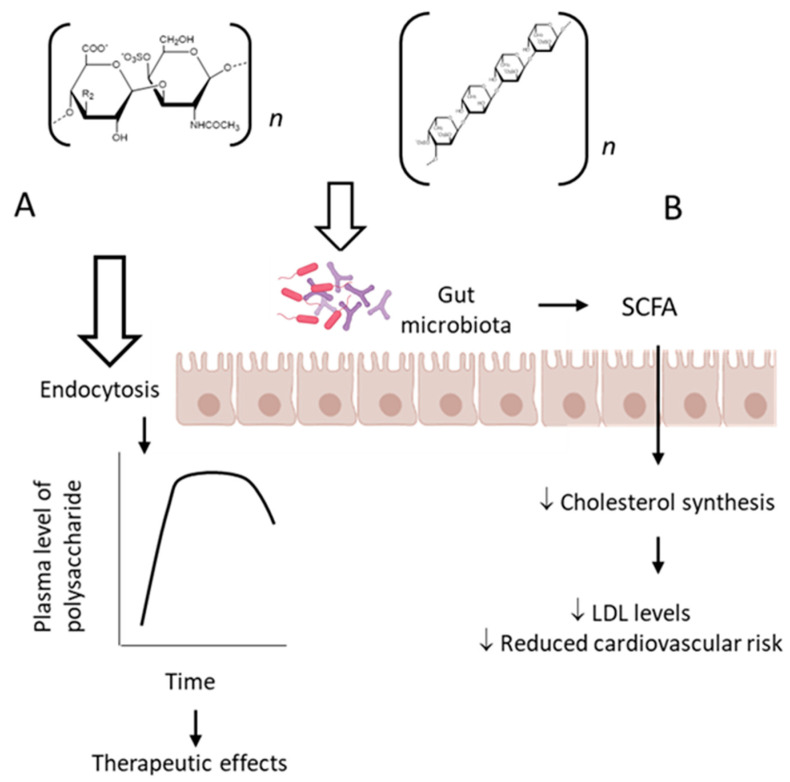
Pharmacological effects of sulfated polysaccharides after oral administration. (**A**) As the polysaccharides pass through the gastrointestinal tract, they are absorbed, probably by endocytosis due to their high-molecular-weight, and reach the bloodstream. Subsequently, they are distributed among various tissues and excreted unchanged and/or metabolized, as classically described for orally active drugs. (**B**) Alternatively, polysaccharides can exert a prebiotic effect by modulating the intestinal microbiota, which will produce short-chain fatty acids that can pass the intestinal mucosa by passive diffusion and reach the bloodstream, inhibiting cholesterol synthesis and reducing cardiovascular risk.

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
