# Peer review of "Pharmacological Activities of Sulfated Fucose-Rich Polysaccharides after Oral Administration: Perspectives for the Development of New Carbohydrate-Based Drugs"

_marinedrugs, 2021, doi:10.3390/md19080425_

Round 1

Reviewer 1 Report

In this manuscript the  Authors review recent literature regarding the potential therapeutic effects of sulfated fucose rich polysaccharides in several human pathologies. The topic is well addressed, organized and comprehensible also for not expert reader.

Author Response

This reviewer had no suggestion for modification of the manuscript.

Reviewer 2 Report

Comments and suggestions for authors:

This review summarizes the therapeutic effects achieved after oral administration of the sulfated fucose-rich polysaccharides in various pathological processes. The manuscript is well written with comprehensive information and clear description, but I still have some questions as follows:

  1. The literature collected from the current manuscript shows that oral high-molecular-weight fucoidan seems to have more benefits than low-molecular-weight, but generally, the absorption rate of small-molecular-weight should be better. Why is there a contradiction?
  2. Is it possible to supplement the criteria for distinguishing the high molecular weight and low molecular weight of fucoidan?
  3. As the author said, due to individual differences between species and strain levels, the metabolic ability of the microbiota to hydrolyze molecules is different. However, from the author's point of view, does the source of polysaccharides also affect the growth of intestinal microbes? In other words, which sources of sulfated polysaccharides are more effective in establishing a good intestinal bacterial environment?
  4. There seems to be an incorrect text on line 510, please confirm and correct it.

Author Response

Response to Reviewer 2 Comments

Point 1: The literature collected from the current manuscript shows that oral high-molecular-weight fucoidan seems to have more benefits than low-molecular-weight, but generally, the absorption rate of small-molecular-weight should be better. Why is there a contradiction?

Response 1: In fact, the absorption rate of small-molecular-weight is better than high-molecular-weight molecules. However, the pharmacological activities of sulfated fucose-rich polysaccharides depend on their molecular weight as well.

For fucosylated chondroitin sulfate, for example, some papers showed that reduction of molecular weight abolishes its anticoagulant activities (doi: 10.1097/00001721-200401000-00008; doi: 10.1016/j.carbpol.2013.07.063, doi: 10.1016/j.ejmech.2017.07.065).

The extent of the pharmacological effect after oral administration depends on a balance between the molecular weight and the rate of absorption of the polysaccharide into the bloodstream.

Point 2: Is it possible to supplement the criteria for distinguishing the high-molecular-weight and low-molecular-weight of fucoidan?

Response 2: We added two new Tables summarizing the structural characteristics of fucoidan and fucosylated chondroitin sulfate, including the molecular weight (Table 1 and Table 2, RM).

Point 3: As the author said, due to individual differences between species and strain levels, the metabolic ability of the microbiota to hydrolyze molecules is different. However, from the author's point of view, does the source of polysaccharides also affect the growth of intestinal microbes? In other words, which sources of sulfated polysaccharides are more effective in establishing a good intestinal bacterial environment?

Response 3: We agree that the source of polysaccharide may differ in the ability to modulate gut microbiota. The prebiotic function of sulfated polysaccharides is a new field of study and the first reports were publicated only recently.

Obviously, it also may depend on the polysaccharide structure. Further work is necessary to assess whether these changes in both bacterial composition and sulfated polysaccharide degradation also occur in vivo, and critically, whether such changes are responsible for some of the biological effect of these sulfated fucose-rich polysaccharides after oral administration (pg 22, line 524-27, RM).”

 The consumption of marine foods may also be another relevant mechanism for the oral absorption of sulfated polysaccharides (pg 21, line 521, doi:10.3390/md16080254).

Point 4: There seems to be an incorrect text on line 510, please confirm and correct it.

Response 4: We corrected this sentence (pg 20, line 462, RM) and also fixed grammar and spelling errors in the manuscript.

Reviewer 3 Report

Fonseca and Mourão reviewed the pharmacological activities of sulfated fucose-rich polysaccharides after oral administration. After close evaluation of the manuscript I suggest revision according to next points:

  1. In Section 2: the inhibition of DPP-IV is one of the possible mechanisms involved in the anti-hyperglycemic activity of fucoidan (see a recent paper about Mechanisms of Bioactivities of Fucoidan from the Brown Seaweed Fucus vesiculosus). 
  2. In Tables 1-5: In column "Polysaccharide" I recommend providing more data about polysaccharides (MW, sulfate content, structural sugar units, etc). In column "Major observations and mechanism proposed" I suggest providing real results (the phrase "...Decreased plasma level of lipids" is less informative than "Decrease plasma level of lipids by ...%"; "Inhibition of HMG-CoA reductase" - please indicate IC50 or some more exact details, etc).
  3. In section 6: extra information could be found in the above-mentioned paper (Mechanisms of Bioactivities of Fucoidan from the Brown Seaweed Fucus vesiculosus).
  4. In section 7: pharmacokinetic of fucoidan is discussed in several other papers: https://doi.org/10.1016/j.ijbiomac.2019.10.018; https://doi.org/10.3390/md18110557; https://doi.org/10.3177/jnsv.63.419; https://doi.org/10.3390/md16040132; 
  5. https://doi.org/10.1358/mf.2005.27.10.948919
  6. The clinical data are very important for the implementation of polysaccharides used in practical medicine. Therefore I suggest summarising all clinical data in a special section and discuss it more precisely with the focus on practical; application. The clinical data should be addressed in the conclusion as well.
  7. In Conclusion: the authors provide the statement "Nanoparticulate delivery systems have been extensively in-592 vestigated as oral delivery vehicles..." however this aspect was not discussed in the paper.

Author Response

Response to Reviewer 3 Comments

Point 1: In Section 2: the inhibition of DPP-IV is one of the possible mechanisms involved in the anti-hyperglycemic activity of fucoidan (see a recent paper about Mechanisms of Bioactivities of Fucoidan from the Brown Seaweed Fucus vesiculosus). 

Response 1: We added a sentence including the inhibition of DPP-IV as one of the possible mechanisms involved in the hypoglicemic activity of fucoidan. (pg 15, lines 306-07, RM).

Point 2: In Tables 1-5: In column "Polysaccharide" I recommend providing more data about polysaccharides (MW, sulfate content, structural sugar units, etc). In column "Major observations and mechanism proposed" I suggest providing real results (the phrase "...Decreased plasma level of lipids" is less informative than "Decrease plasma level of lipids by ...%"; "Inhibition of HMG-CoA reductase" - please indicate IC50 or some more exact details, etc).

Response 2: We added two new Tables summarizing the structural characteristics of fucoidan and fucosylated chondroitin sulfate, including the molecular weight (new Tables 1 and 2, RM). We also included more details about the % of alteration in some pharmacological activities on modified Tables 3, 6 and 7, RM. However, we choose not to go into numeric detail on all reported pharmacological activities to avoid overloading information and give the readers a general view of the mechanisms proposed by the authors.

Point 3: In section 6: extra information could be found in the above-mentioned paper (Mechanisms of Bioactivities of Fucoidan from the Brown Seaweed Fucus vesiculosus).

Response 3: In this paper, the authors showed a prolongation of aPTT and TT values but did not show any specific mechanism associated with the anticoagulant activity of fucoidan.

Thus, fucoidan from F. vesiculosus showed potent anticoagulant activities. It prolongs APTT and TT significantly and concentration-dependently and prolongs PT at high concentrations. This showed that the studied fucoidan may have an effect on intrinsic/common pathways and little effect on the extrinsic mechanism.”

The ability to prolong aPTT, TT or PT values is due to the sensitivity of the reagents used in these assays when incubated with increasing concentrations of the polysaccharides. Heparin prolongs aPTT and TT values, but shows a weak correlation with PT values. Likewise, low-molecular-weight heparin affects mostly Xa activity, as monitored using chromogenic assays.  

Point 4:  In section 7: pharmacokinetic of fucoidan is discussed in several other papers: https://doi.org/10.1016/j.ijbiomac.2019.10.018; https://doi.org/10.3390/md18110557; https://doi.org/10.3177/jnsv.63.419; https://doi.org/10.3390/md16040132; 

https://doi.org/10.1358/mf.2005.27.10.948919

Response 4: We added some sentences in the manuscript regarding the pharmacokinetics of fucoidan discussed in these papers. (pg 20, lines 477-78; pg 21 lines 486-490, 521-22; RM).

Point 5: The clinical data are very important for the implementation of polysaccharides used in practical medicine. Therefore, I suggest summarising all clinical data in a special section and discuss it more precisely with the focus on practical; application. The clinical data should be addressed in the conclusion as well.

Response 5: We summarized all clinical data in section 7 of the RM and addressed this point in the conclusion as well. (pg 19, line 407-40, RM).

Point 6:  In Conclusion: the authors provide the statement "Nanoparticulate delivery systems have been extensively in-592 investigated as oral delivery vehicles..." however this aspect was not discussed in the paper.

Response 6: We deleted this sentence since this aspect was not discussed in the paper.

Round 2

Reviewer 3 Report

The manuscript was improved, however some inportant points are still not addressed correctly^

  1. The real results are iportant for review papers. It is important to understand how significant was for example "Improvement of reverse cholesterol transport..." for 5% or 25% or more. "Inhibition of HMG-CoA reductase///" please provide IC50 (if available) (Table 3)? "Inhibition of Lymphokine-activated killer T-cell-originated
    protein kinase..." IC50? (Table 4/ tnc/
  2. Authors have not cited all suggested papers. For example https://doi.org/10.3390/md18110557

Author Response

1) We added to the revised manuscript (RM) the quantitative data of the pharmacological activities available in the literature (see modified tables of the RM).

2) The reference mentioned by the reviewer  was added to the RM (new reference 84, see lines 486/487).

Round 3

Reviewer 3 Report

Authors adequately addressed my recommendations.